# Intertidal resource use over millennia enhances forest productivity

Andrew J. Trant[1,2,3], Wiebe Nijland[2,3,4], Kira M. Hoffman[2,3], Darcy L. Mathews[2,3], Duncan McLaren[2,5], Trisalyn A. Nelson[2,6] & Brian M. Starzomski[2,3]

Human occupation is usually associated with degraded landscapes but 13,000 years of repeated occupation by British Columbia's coastal First Nations has had the opposite effect, enhancing temperate rainforest productivity. This is particularly the case over the last 6,000 years when intensified intertidal shellfish usage resulted in the accumulation of substantial shell middens. We show that soils at habitation sites are higher in calcium and phosphorous. Both of these are limiting factors in coastal temperate rainforests. Western redcedar (*Thuja plicata*) trees growing on the middens were found to be taller, have higher wood calcium, greater radial growth and exhibit less top die-back. Coastal British Columbia is the first known example of long-term intertidal resource use enhancing forest productivity and we expect this pattern to occur at archaeological sites along coastlines globally.

[1] School of Environment, Resources and Sustainability, University of Waterloo, 200 University Avenue West, Waterloo, Ontario, Canada N2L 3G1. [2] Hakai Institute, Calvert Island, PO Box 309, Heriot Bay, British Columbia, Canada BC V0P 1H0. [3] School of Environmental Studies, University of Victoria, PO Box 1700 STN CSC, Victoria, British Columbia, Canada V8W 2Y2. [4] Department of Geography, University of Victoria, PO Box 1700 STN CSC, Victoria, British Columbia, Canada V8W 2Y2. [5] Department of Anthropology, University of Victoria, PO Box 1700 STN CSC, Victoria, British Columbia, Canada V8W 2Y2. [6] School of Geographical Sciences and Urban Planning, Arizona State University, PO Box 875302, Tempe, Arizona 85287-5302, USA. Correspondence and requests for materials should be addressed to A.J.T. (email: atrant@uwaterloo.ca).

Humans are usually associated with degraded landscapes and the 'forest primeval' is thought to be free of human influence. Removal of habitat, depletion of soil nutrients, forest productivity declines and reduction in biodiversity are commonly cited outcomes of modern human land-use patterns[1,2]. There is growing evidence, however, of long-term land and resource-use practices such as soil terracing[3], fertilization[4], forest management[5,6] and controlled burning[7], which provide alternative models to human presence resulting in environmental degradation. Here we show that millennia of resource use by coastal First Nations in British Columbia, Canada, contributes significant marine-derived nutrients and alters site conditions that enhance temperate rainforest productivity. We focus on western redcedar (*Thuja plicata* Donn ex D. Don), a species of cultural importance throughout the Pacific Northwest of North America[8,9], and a valuable timber tree[10].

Coastal First Nations of British Columbia have a long and vibrant history, with oral and archaeological data providing evidence of over 13,000 years of repeated occupation. Long-term records of occupation are found on the Central Coast of British Columbia, as sea level has been relatively static throughout the Holocene[11,12]. The term 'habitation site' refers to a place where people lived in the past resulting in the accumulation of a physical-material record following years of seasonal or multi-seasonal occupation. Most habitation sites are situated near the shoreline, where people lived in proximity to a diversity of terrestrial (for example, root gardens, managed plant resources and mammals) and especially marine (for example, clams and fish) resources[13–15]. Significantly, these resource-use patterns resulted in the importation of shell from the intertidal zone into

the terrestrial environment. Typically, the major constituents of habitation site shell middens are shell, rock, bone, charcoal, plant remains (mainly rootlets), artefacts, archaeological features and organic soil.

Disposal and stockpiling of shell, as well as the cultural use of fire, altered the species composition of the forest and understory in and around habitation sites[16,17]. The legacy of human use is most visibly preserved in the accumulation of substantial shell middens, which in some cases exceed 5 m in depth[18] and cover thousands of square meters of forest area as the shells are placed for terracing and drainage or are simply discarded as refuse. Less visible legacies at shell midden sites are the effects of ancient fires in the form of charcoal from repeatedly buried hearths[19] and site-level burning before reoccupation[20], which increases nutrient availability[21] and shifts competition[22]. Although these sites continue to have important cultural value, many have not been occupied with the same intensity or regularity for at least the last 150–250 years, especially since smallpox epidemics in the nineteenth century[23,24]. However, the western redcedars that have subsequently grown on these shell middens have been intensively culturally modified for harvesting wood and bark. The vast majority of the study area has not been commercially logged or developed, helping to preserve shell middens, an abundance of culturally modified trees, and many stone fish traps and clam gardens in adjacent intertidal areas[25,26].

The long-term human harvesting and deposition of shellfish and other animal remnants from the marine to near-shore environment represents a significant marine-derived nutrient input and modifier of soil pH. Some marine-derived nutrients associated with these sites, such as nitrogen from salmon carcasses moved inland by predators and scavengers such as bears and wolves, have also

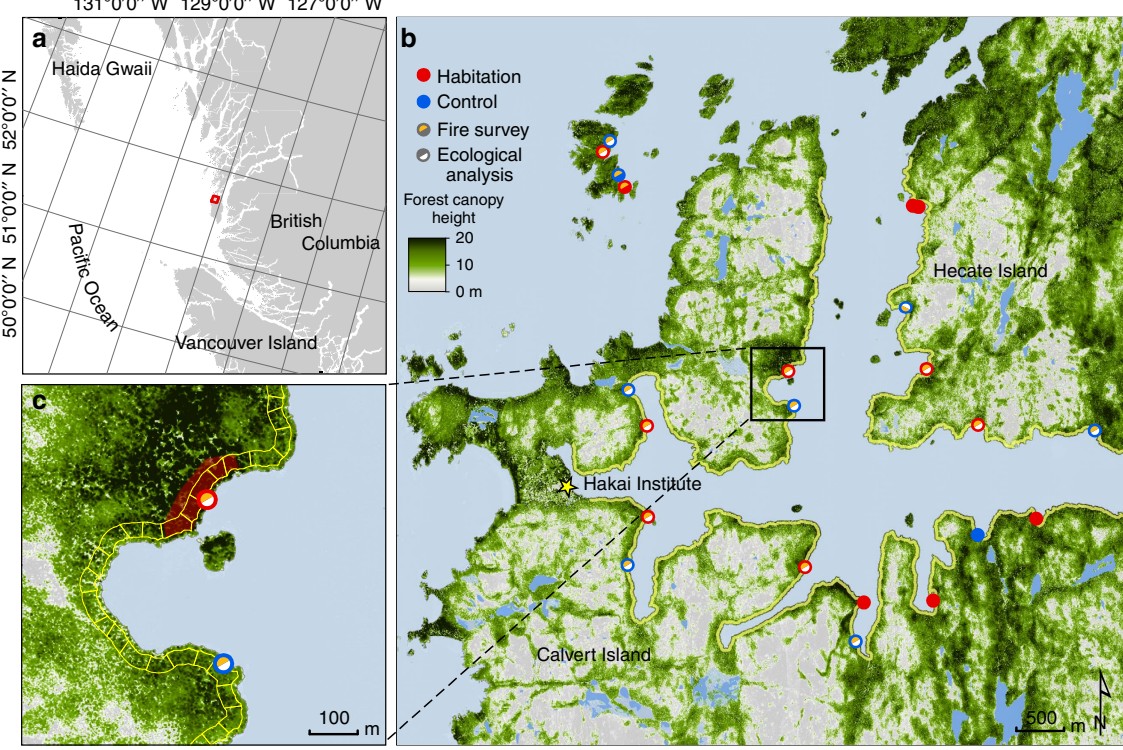

**Figure 1 | Study area map.** The study area (**a**) includes northwest Calvert Island and southwest Hecate Island on the Central Coast of British Columbia, Canada. (**b**) Map of the study area with all habitation sites (red) and control sites (blue). Fire surveys (orange-filled red circles, $n = 7$) and detailed ecological analyses (white-filled red circles, $n = 6$) were carried out on a subset of habitation sites. (**c**) Close-up map of habitation site (red circle) with approximate shell midden delineated by transparent red polygon. The blue circle shows the control site. Cells ($50 \times 30$ m) used for extracting lidar data are shown in yellow. Data for base maps from https://catalogue.data.gov.bc.ca/dataset/nts-bc-coastline-polygons-1-250-000-digital-baseline-mapping-nts used with permission under http://open.canada.ca/en/open-government-licence-Canada.

been demonstrated to be important to community and ecosystem structure[27], although these nutrients are expected to deplete relatively quickly[28]. However, the slow release of calcium from degrading shells ($CaCO_3$) is more persistent[29]. Calcium deficiencies are common in forested ecosystems and have been proposed as a significant contributor to top die-back in western redcedar[30], although see ref. 31, a condition that results in the death of tree crowns and could represent a significant economic loss to forestry. Further, the addition of $CaCO_3$ and charcoal from fire increases soil pH, which in turn increases the availability of macronutrients such as phosphorous[32]. Shell midden, as a bulk sedimentary matrix, changes the physical structure of the soil by improving drainage[33] while the deposition of charcoal increases the porosity of highly weathered soils and also increases phosphorous[34]. Increased soil pH and drainage contribute to enhanced forest productivity.

Using a combination of airborne lidar and field-based ecological methods, we examine how the consequences of long-term site occupation, such as shell middens and fire, influence measures of forest productivity at our study site on the Central Coast (Fig. 1a,b). Forest productivity, expressed by forest canopy height, forest width, vegetation greenness and forest canopy cover (that is, productive forests extending further inland) and vegetation greenness at habitation sites was compared with forests along the entire coast contained within our study area using lidar extracted in $30 \times 50$ m cells set in from the coastal forest boundary by 10 m (Fig. 1c).

## Results

### Forest productivity is highest near and on human habitation sites.

To explain and understand differences in forest productivity, we obtained eco-cultural parameters (shell midden depth, prevalence of fire, soil nutrients, wood nutrients and individual tree growth metrics) using a paired design with control sites located along the coast in areas with similar forest composition and high productivity, but lacking in shell middens (that is, these sites are assumed not to be habitation sites and archaeological surveys confirm this).

Measures of forest productivity were highest in close proximity to habitation sites and decreased within ∼200 m from the shell midden boundary. Trees growing on habitation sites were taller than those growing off of habitation sites (Welch's two sample $t$-test: $t = 2.61$, d.f. $= 12.27$, $P = 0.022$; Fig. 2a) and deeper shell middens exhibited a larger effect (Fig. 2b). This pattern was also observed in forest width ($t = 3.34$, d.f. $= 12.36$, $P = 0.006$; Fig. 3) and vegetation greenness ($t = 3.12$, $P = 0.009$; Fig. 4), and, to a lesser extent, forest canopy cover ($t = 1.86$, d.f. $= 12.58$, $P = 0.087$; Fig. 5). Only the amount of insolation was better than the distance from the shell midden boundary on habitation sites in explaining patterns of forest height across the coastal forests in our study site (28 and 24%, respectively) (Table 2). The most important factor explaining patterns of forest width was distance from shell midden boundary on habitation sites (38%) (Table 2). For vegetation greenness and forest canopy cover, distance from habitation site was similar to slope, aspect and elevation in explaining forest productivity patterns (∼20%) (Table 2).

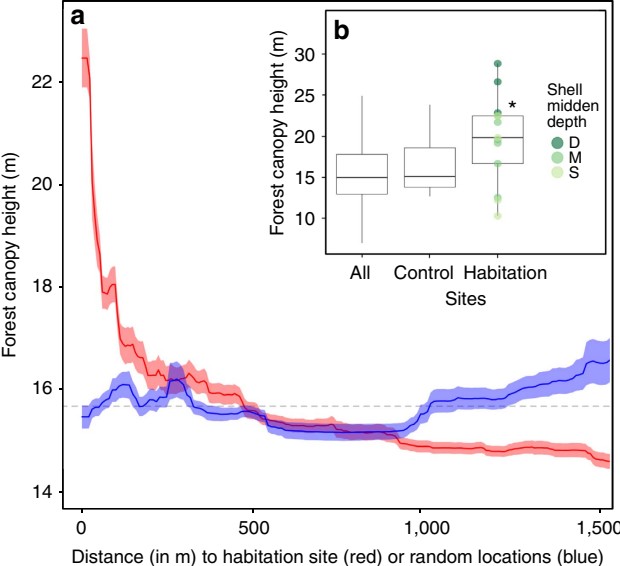

**Figure 2 | Habitation site and shell midden depth influence on forest canopy height.** (**a**) Forest canopy height (in metres) by distance from the shell midden boundary (red) or distance from random locations (blue) shown with s.d. (red and blue ribbon) of bootstrapped boosted regression tree models, which are an additive regression model where terms are simple trees that are fitted in a forward, stagewise manner[52]. Mean canopy height across entire study area is shown by a dashed line (15.70 m). All distances are measured within the 30 m ribbon delineated from lidar data. (**b**) Box plot of difference in canopy height by site type and the depth of shell midden (deep (D) = dark green, medium depth (M) = green, shallow (S) = light green) with forest canopy height at habitation sites significantly greater than all non-habitation sites ($P = 0.02$). Control sites are located along the coast in areas without shell midden where 'All' sites include the entire study area shown in yellow in Fig. 1b. Boxes represent first, second and third quartiles with whiskers extending to the highest and lowest value that is within $1.5 \times$ interquartile range (IQR). Significantly higher values are denoted by '*'.

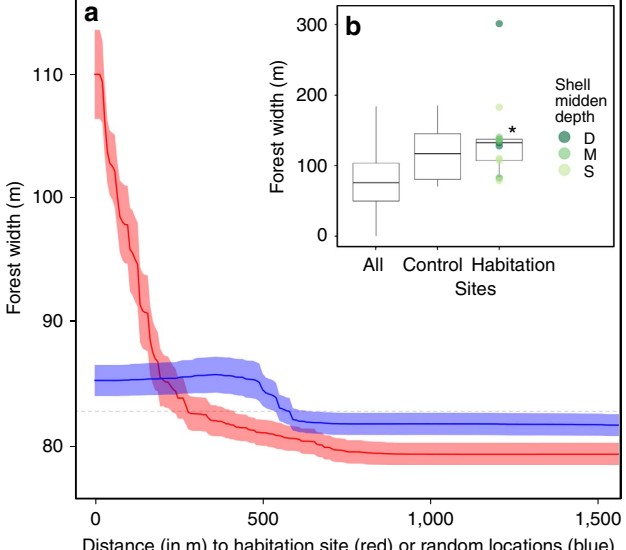

**Figure 3 | Habitation site and shell midden depth influence on forest width.** (**a**) Forest width (in metres) by distance from the shell midden boundary (red) or distance from random locations (blue) shown with s.d. of bootstrapped boosted regression tree models and mean forest width across entire study area shown by a dashed line (82.67 m). (**b**) Box plot of difference in canopy height by site type and size of the depth of shell midden (deep (D) = dark green, medium depth (M) = green, shallow (S) = light green) with forest width at habitation sites being significantly greater than all non-habitation sites ($t = 3.341$, $P = 0.006$). Control sites are located along the coast in areas without shell midden where 'All' sites include the entire study area shown in yellow in Fig. 1. Boxes represent first, second and third quartiles with whiskers extending to the highest and lowest value that is within $1.5 \times$ IQR. Significantly higher values are denoted by '*'.

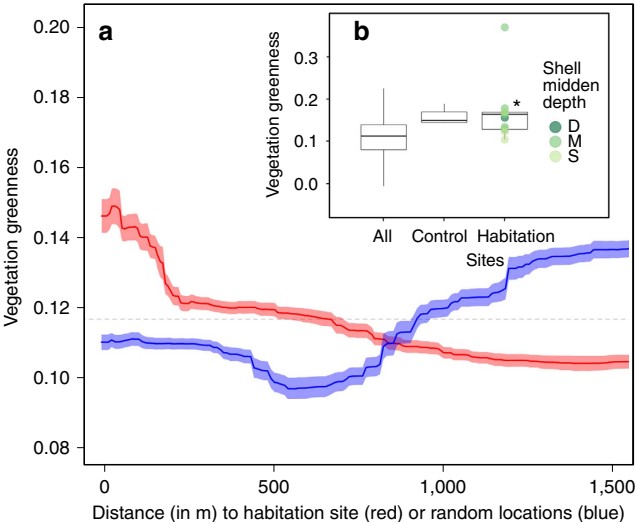

**Figure 4 | Habitation site and shell midden depth influence on vegetation greenness.** (**a**) Vegetation greenness by distance from the shell midden boundary (red) or distance from random locations (blue) shown with s.d. of bootstrapped boosted regression tree models and mean vegetation greenness across entire study area shown by a dashed line (0.117). (**b**) Box plot of difference in vegetation greenness by site type and the depth of shell midden (deep (D) = dark green, medium depth (M) = green, shallow (S) = light green) with vegetation greenness at habitation sites being significantly higher compared with all non-habitation sites ($t = 3.117$, $P = 0.009$). Control sites are located along the coast in areas without shell midden where 'All' sites include the entire study area shown in yellow in Fig. 1. Boxes represent first, second and third quartiles with whiskers extending to the highest and lowest value that is within $1.5 \times$ IQR. Significantly higher values are denoted by '*'.

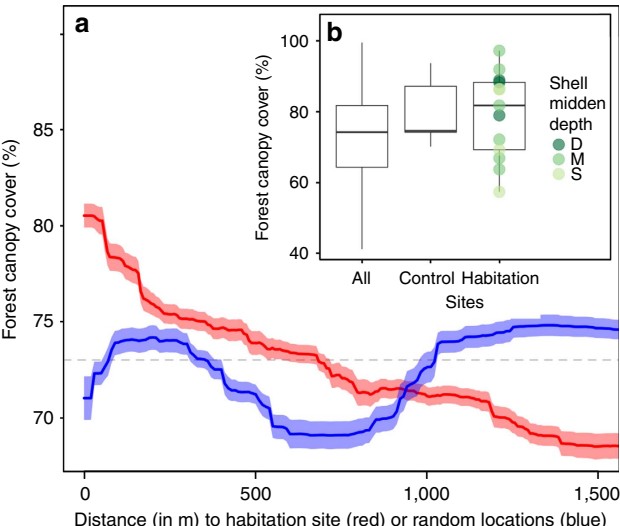

**Figure 5 | Habitation site and shell midden depth influence on forest canopy cover.** (**a**) Forest canopy cover (in percentage) by distance from the shell midden boundary (red) or distance from random locations (blue) shown with s.d. of bootstrapped boosted regression tree models and mean forest canopy cover across entire study area shown by a dashed line (73.01%). (**b**) Box plot of difference in forest canopy cover by site type and the depth of shell midden (deep (D) = dark green, medium depth (M) = green, shallow (S) = light green) with forest canopy cover at habitation sites being moderately higher compared with all non-habitation sites ($t = 1.857$, $P = 0.087$). Control sites are located along the coast in areas without shell midden where 'All' sites include the entire study area shown in yellow in Fig. 1. Boxes represent first, second and third quartiles with whiskers extending to the highest and lowest value that is within $1.5 \times$ IQR.

**Nutrient modifications enhance forest productivity.** Soil from habitation sites can explain these striking differences in productivity, being significantly higher in calcium (nested analysis of variance: $F = 12.34$, d.f. $= 1$, $P = 0.001$; Fig. 6a), phosphorous ($F = 7.01$, d.f. $= 1$, $P = 0.012$) and pH ($F = 19.55$, d.f. $= 1$, $P = 0.001$; Fig. 7a), and lower in potassium ($F = 5.03$, d.f. $= 1$, $P = 0.032$; Fig. 7b). Nitrogen, sulphur and magnesium levels were similar (Fig. 7c–f). Western redcedar trees growing on habitation sites had significantly higher wood calcium levels ($F = 4.41$, d.f. $= 1$, $P = 0.037$; Fig. 6b), greater radial growth ($F = 23.61$, d.f. $= 1$, $P < 0.001$; Fig. 6c) and experienced less top die-back ($\chi^2$-test $= 14.19$, d.f. $= 1$, $P < 0.001$; Fig. 6d). Lastly, above-ground evidence of fire was exclusively associated with habitation sites (Table 2) and this may also have contributed to forest productivity.

## Discussion

Although pre-occupation site productivity is unknown, multiple lines of evidence demonstrate that productivity is enhanced at habitation sites, independent from any site selection effect that may exist.

First, from an ecological and archaeological perspective, the dominant tree species, western redcedar, did not appear in this region until at least 7,000–8,000 Cal BP[35,36], with modern vegetation arising 4,000–2,000 Cal BP[37], millennia after many of these sites were originally occupied (Table 1). Second, shell middens and historic fire dramatically change local site topography, nutrient dynamics, soil pH and site drainage. We are therefore confident that current measures of forest productivity are enhanced by these sites being occupied, regardless of site conditions before occupation.

Local patterns in forest productivity in these coastal regions have previously been shown to be primarily driven by insolation (aspect), water retention of soils and nutrient availability[38]. Our data demonstrate similar effects of these factors, but also reveal that distance from habitation sites is routinely ranked as one of the most important predictors of forest productivity (Table 2). Thus, shell midden habitation sites and the occurrence of historic fires transform the local site conditions through increased soil pH and macronutrient inputs[39,40], better site drainage and more level ground, thus offering a mechanism by which forest productivity is enhanced in an otherwise nutrient-limited ecosystem. Shell middens have been shown to alter soil chemistry resulting in greater vegetation cover and higher species richness[29] with qualitative reports of vegetation growing on shell middens being botanically different[41], and greener and denser[42]. However, this is the first known documentation of changes to forest productivity resulting from long-term intertidal resource use patterns. Wider forests around habitation sites may be related to the nutrient and drainage benefits from shell middens. In addition, the cultural use of fire at habitation sites probably extended beyond the boundaries of the shell midden either intentionally or accidentally. Less pronounced effects of habitation sites on forest canopy cover might be influenced more by canopy competition for light than nutrients and site drainage.

We reveal a strong association between habitation sites and low-severity fire. The cool climate, high levels of annual precipitation ($> 4,000$ mm) and low probability of summer lightning on Canada's west coast render lightning-caused fires unlikely[43]. Historically, low-severity fires may have been an important contributor to increased forest productivity in our study area and may also be associated with active management

**Table 1 | Description of habitation and control sites.**

| Island | Site code | Latitude | Longitude | Lidar (L), fire survey (F) and ecological analyses (E) | Midden depth (cm) | Depth to midden (cm) | Continuous habitation | Terminal shell midden age range (1 Sigma Cal AD/BC) | Evidence of fire |
|---|---|---|---|---|---|---|---|---|---|
| Calvert | EjTa15 | 51°39'39 N | 128°07'09 W | L/F/E | 120 | 15 | No | AD 1699–1915* | Yes |
| Calvert | Control | 51°39'35 N | 128°07'18 W | L/F/E | NA | NA | NA | NA | No |
| Calvert | EjTa14 | 51°39'05 N | 128°07'09 W | L/F/E | 300 | 58 | Yes | AD 1325–1344† | Yes |
| Calvert | Control | 51°38'49 N | 128°07'16 W | L/F/E | NA | NA | NA | NA | No |
| Calvert | EjTa04 | 51°39'54 N | 128°05'53 W | L/F/E | 800 | 53 | Yes | AD 1524–1643‡ | Yes |
| Calvert | Control | 51°39'42 N | 128°05'51 W | L/F/E | NA | NA | NA | NA | No |
| Hecate | EjTa19 | 51°39'35 N | 128°04'11 W | L/F/E | 120 | 50 | Unclear | AD 1247–1274§ | Yes |
| Hecate | Control | 51°39'31 N | 128°03'08 W | L/F/E | NA | NA | NA | NA | No |
| Hecate | EjTa13 | 51°39'53 N | 128°04'37 W | L/F/E | 400 | NA | Yes | AD 196–410† | Yes |
| Hecate | Control | 51°40'15 N | 128°04'50 W | L/F/E | NA | NA | NA | NA | No |
| Starfish | EkTa37 | 51°40'59 N | 128°07'21 W | F | 200 | NA | NA | NA | Yes |
| Starfish | Control | 51°40'53 N | 128°07'13 W | F | NA | NA | NA | NA | No |
| Starfish | EkTa38 | 51°41'08 N | 128°07'29 W | F/E | 250 | 100 | NA | BC 1381–1413† | Yes |
| Starfish | Control | 51°41'11 N | 127°07'23 W | F/E | NA | NA | NA | NA | No |
| Calvert | EjTa02 | 51°38'35 N | 128°05'14 W | L | 200 | NA | NA | NA | NA |
| Calvert | EjTa18 | 51°39'2 N | 128°03'38 W | L | 50 | NA | NA | NA | NA |
| Calvert | EjTa06 | 51°38'41 N | 128°04'36 W | L | 50 | NA | NA | NA | NA |
| Calvert | EjTa16 | 51°39'50 N | 128°07'03 W | L | NA | NA | NA | NA | NA |
| Calvert | EjTa17 | 51°38'36 N | 128°06'00 W | L | 150 | NA | NA | AD 1146–1027†,‡ | NA |
| Calvert | EjTa25 | 51°39'44 N | 128°06'59 W | L | NA | NA | NA | NA | NA |
| Calvert | EjTa26 | 51°38'40 N | 128°05'22 W | L | NA | NA | NA | NA | NA |
| Calvert | EkTa02 | 51°40'47 N | 128°04'38 W | L | 50 | NA | NA | NA | NA |
| Calvert | EkTa42 | 51°40'47 N | 128°04'45 W | L | 50 | NA | NA | AD 1151–1032† | NA |

NA, not applicable.
Shell midden depths were obtained using JMC Sub soil probe/ percussion corer. Depth to shell middens were acquired with soil sampling augers and are used to show the amount of soil accumulation on habitation sites as a proxy for the time since these sites were occupied. Continuous habitation refers to whether or not archaeological records at each site suggest continuous seasonal use from time of occupation. Time since occupation is an age estimate from the top of the shell midden using radiocarbon dates (Calibrated years AD/BC). Evidence of fire corresponds to fire surveys at each site where above-ground fire, was documented.
*Reported in McLaren[12].
†Mathews (unpublished).
‡Rahemtulla[18].
§UCIAMS 163735.

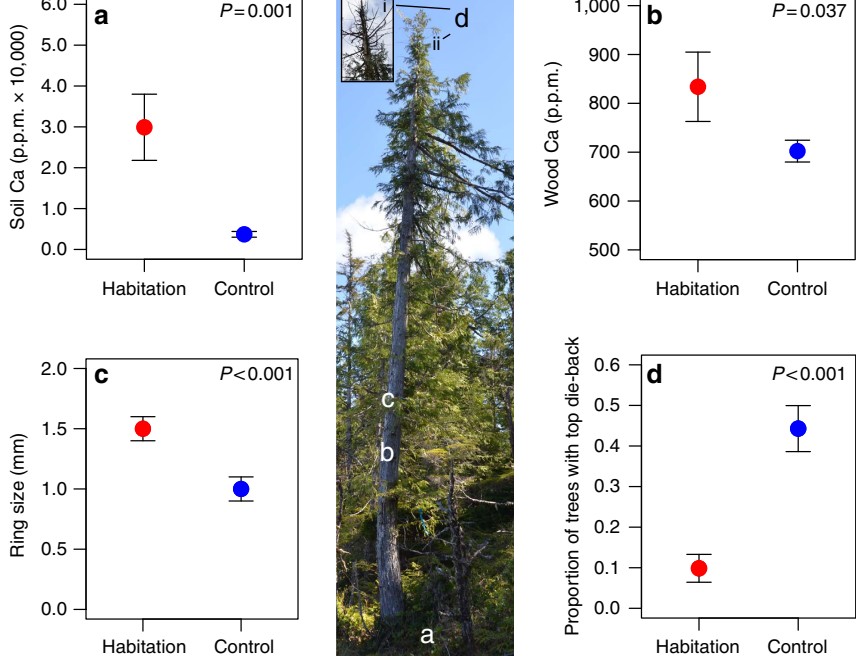

**Figure 6 | Soil calcium dynamics and ecological responses of western redcedar.** Levels of calcium in (**a**) soil and (**b**) western redcedar (*T. plicata*) wood, and with forest productivity metrics of (**c**) radial growth and (**d**) the proportion of western redcedar trees exhibiting top die-back (image showing redcedar with (i) and without (ii) top die-back). Red circles correspond to habitation sites with shell middens (*n* = 6; see Table 1 for details). Blue circles correspond to control sites that were selected based on similar composition (dominance of western redcedar), high productivity and lacking a shell midden (hence was considered not to be a habitation site). Coloured circles are mean values and error bars represent standard error. Image credit: A. Trant.

**Remote sensing data.** Discrete return airborne scanning lidar was acquired on 15 August 2012 by Terra Remote Sensing Inc. (Sidney, BC, Canada). Lidar was collected from 1,150 m AGL at 100 kHz with a maximum scan angle of 26°. The resulting point data have an average point density of 2 points per m$^2$ with an average vertical accuracy of 15 cm. A digital terrain surface was generated from classified ground returns using triangular irregular network interpolation and rasterized at a spatial resolution of 1.0 m. The terrain model was used to derive elevation, slope, aspect and upstream area layers, as well as to normalize non-ground lidar returns to height above ground surface. Vegetation greenness as expressed in the Normalized Difference Vegetation Index was calculated from SPOT6 images acquired on 10 August 2014 and corrected to surface reflectance.

**Boosted regression tree modelling.** To evaluate the effect of proximity to habitation sites on vegetation structure compared with other landscape factors we fit a Boosted Regression Tree model for four metrics of forest productivity acquired through remote sensing: forest canopy height, forest width, vegetation greenness, represented by Normalized Difference Vegetation Index, and forest canopy cover. Boosted Regression Tree modelling is flexible, as it accepts both continuous and classified variables and also allows for nonlinear relations[52]. In addition to the terrain covariates, we also included surface material and coastal morphology and exposure from the British Columbia shorezone map[53]. The study area has a strong gradient of forest structure perpendicular to the shoreline with productive forests on the coastline and mostly marginally productive vegetation and bog areas inland. To account for this gradient and coastal edge effects, we extracted all vegetation data and the covariates from cells that extended 30 m inland and ran 50 m along the shoreline. These cells were set in 10 m from the coastal forest boundary given that vegetation structure is highly variable due to exposure and edge effects. All model data were averaged to 50 m wide sections of this coastal buffer on the study area ($n = 600$). Models were built using Boosted Regression Trees from the 'gbm' package in R statistical software[54], following the methods described in Friedman[55,56]. We determined optimal fitting parameters in an initial exploration using tenfold cross-validation at a tree complexity of 4, minimum number of samples per node at 5 and a learning rate of 0.01. The number of trees was optimized for each individual model and was between 700 and 1,400. The only model settings and covariates that varied were distance to habitation and control sites, and we focus on the effect of these parameters. Model performance is indicated by the per cent deviance explained and we consider the effect of individual covariates as their relative influence (as per cent of total deviance explained). Models were bootstrapped (100 runs using 80% of data per run) to generate the s.d. that appears on mean values by distance from habitation site and random location. Differences in forest canopy height, forest width, forest canopy cover and vegetation greenness between habitation sites and non-habitation sites were assessed using a Welch's two-sample $t$-test.

**Fire surveys.** We ran belt transects to examine the presence of fire in forests surrounding habitation and control sites ($n = 7$). The location of transects was randomly assigned proportional to the size of the nearshore site area and transects 6 m wide and 30 m long ($n = 3$) were completed perpendicular to the shoreline at each site. Trees with a diameter at breast height $> 15$ cm diameter were sampled and tree height, health (top die-back, rot) and decay class were recorded. We also recorded the presence of culturally modified trees, char and fire scars.

**Nutrients and tree radial growth.** At each habitation and control site ($n = 6$), we sampled 9–14 western redcedar trees and using a 5.15 mm increment borer, extracted 2 cores at breast height (1.4 m) for determining radial growth and wood nutrient analyses. For each tree, we recorded physical attributes of diameter at breast height, height and health (for example, top die-back). Increment samples for radial growth analyses were air-dried, mounted and sanded to a high polish. All western redcedar samples were measured and counted using a Velmex sliding stage micrometre (precision 0.001 mm). For wood nutrient analyses, increment samples were cut into decadal units using a scalpel under a dissection scope and stored in paper coin envelopes. Three samples from each tree were analysed for macro and micronutrients for the period 1905–1914, 1955–1964 and 2005–2014, representing evenly spaced decadal sections across the longest shared record.

At all habitation and control sites, we collected soil samples using an AMS soil auger with a diameter of 5.7 cm. Soil samples were taken at 30 cm below litter layer where we were assumed to be in the active rooting layer and when shell middens were present, samples were taken from soils developed on top of shell middens. Large roots and plant material were removed from the samples. Samples were stored in plastic bags and refrigerated until analysed. All soil and wood samples were analysed for nutrient content by the North Road Laboratory facility of the Research, Innovation and Knowledge Management Branch of the British Columbia Ministry of Forest, Lands and Natural Resources.

Differences in nutrients and radial growth were determined using a nested analysis of variance with sites being nested within site type (habitation or control). Differences in top die-back between site type were determined using a $\chi^2$-test to account for binary data. To be consistent between approaches, five trees were excluded from analyses, because they were $> 40$ m from the shoreline and hence were outside of the area included in lidar data.

**Data availability.** The data that support the findings of this study are available from the corresponding author upon request.

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

## Acknowledgements

This research was supported by funding to B.M.S from the Hakai Institute, an NSERC Discovery Grant (386594-2010), the Canada Foundation for Innovation Leader's Opportunity Fund (26102), the British Columbia Knowledge Development Fund (804609) and the Ian McTaggart Cowan Professorship at the University of Victoria. We thank E. Peterson, C. Munck, B.C. Parks, J. Bennett, C. Darimont, C. Dawson, J. Fisher, O. Fitzpatrick, K. Fretwell, S. Friesen, B. Hawkins, W. Housty, P. Johnson, K. Jordan, A. MacKinnon, J. Maxwell, W. McInnes, I. McKechnie, M. Oelbermann, F. Rahemtulla, J. Reynolds, N. Shackelford, J. Silberg, D. Smith, J. Stafford, N. Turner, E. Urquhart, J. Vickers, E. White and S. Wickham.

## Author contributions

A.J.T. and B.M.S. originally conceived the study, with W.N. and K.M.H. adding additional methods. A.J.T., K.M.H., B.M.S., D.M. and D.L.M. conducted the fieldwork. A.J.T. and W.N. performed the data analysis aided by B.M.S. and T.A.N. A.J.T., B.M.S., K.M.H. and W.N. drafted the manuscript. All authors discussed the results and commented on the manuscript.

## Additional information

**Competing financial interests:** The authors declare no competing financial interests.

**How to cite this article**: Trant, A. J. *et al.* Intertidal resource use over millennia enhances forest productivity. *Nat. Commun.* 7:12491 doi: 10.1038/ncomms12491 (2016).

