## [Peer Review File · Nature Communications]

Reviewers' comments:

Reviewer #1 (Remarks to the Author):

Summary: The manuscript in review provides an interesting side-story about historical human inputs of calcium via accumulation of shell middens and the slow degradation that provided a steady leaching of Ca, which reduced the Ca deficiency otherwise seen in these soils. The normal Ca depletion has led to die-back of the tops of tall trees in the region, but not where the deficiency was counteracted by the slow additions of Ca from the middens. This story is interesting in that it reminds us to understand the historic land use effects on current forest structure and function, particularly in long-lived forests like the Coast Range of western N America.

Originality: The paper isn't original in terms of the N-Ca coupling, but it has an interesting exception to the rule when historic coastal human habitation alters this coupling via effects on Ca. It is well-known that high nitrogen (N) accumulation in Coast Range forests of Oregon, Washington and British Columbia acidified soils and depleted calcium. A recent study determined that biotic conservation and tight recycling of Ca increased in response to whole-ecosystem Ca depletion, which was indicated by preferential accumulation of Ca in biomass and surface soil. i.e. coupled N-Ca cycling under long-term soil N enrichment.

Methods, statistics: The study design and statistics are appropriate, with paired plots (with and without shell middens), which resulted in finding taller canopies, higher productivity, and less top die-back associated with middens. In other studies, Ca pools in aboveground biomass decreased significantly as soil N increased (e.g. Perakis et al. 2013), whereas this new study shows that western redcedar grown on middens showed significantly higher wood calcium levels. That is, the results are consistent with previous studies along nitrogen gradients in the Coast Range.

Conclusions, clarity and content: The paper stands alone on this topic. The justification for the importance of this finding is weak. I don't think it is useful to suggest that this study challenges the assumption that human activities degrade forests, such as by commercial logging and other resource extraction. These are completely different activities, and they have been shown to degrade forests. It sounds as if the authors are trying to say that if the middens are good for forests, so are all of these other activities, when I don't think that is their intent. Rather, simply state that historical human activity in coastal areas can result in pockets of higher forest productivity associated with the slow leaching of shell middens and fish bones than might otherwise be predicted for high soil N and depleted Ca. Coastal areas, rivers and lakes are obvious places where historic human habitation and consumption of fish and shellfish could leave their marks on the soil fertility and forest productivity that are different from the surrounding forests. The paper is publishable in Nature once revisions improve this part of the paper and the citations.

Suggested Citation: Perakis, S. et al. 2013. Forest calcium depletion and biotic retention along a soil nitrogen gradient. *Ecological Applications* 23:1947-61.

Reviewer #2 (Remarks to the Author):

Summary of key results

The author tests in this study the effect of large archaeological shell middens -accumulated in the coast of British Columbia during the last 6,000 years as a result of shellfish harvesting by the region's First Nations- on the growth of native red cedar forests. Because this region is very wet, with more

than 4,000 mm of rainfall that leach cations away from the acidic local soils, calcium is a limiting resource for plant growth. The authors show that red cedar trees established on large archaeological shell middens significantly enhance their growth and increase the productivity of the temperate rainforest.

Originality and interest

The most interesting aspect of this study is the finding that long-term occupation by humans and large scale accumulation of human-created rubble in the form of shell middens is actually contributing in present times to significantly increase the productivity of the coastal temperate rainforest. This finding, the author argues, contradicts the generalized notion that long-term human occupation leads to the formation of degraded landscapes.

Other studies have attempted to evaluate the impact of archaeological middens on current vegetation.

In 1937, Hrdlička (*Science* 86: 559-560) reported that shell middens in Alaska "present wide and in cases seemingly almost absolute botanical differences from the rest of their region." In 1938, Meigs (*Science* 87: 346) noted that middens in Lower California harbored different species and looked greener and denser than the surrounding desert vegetation. Many archaeological studies have been done on the shell middens of the Pacific coast of the Americas, and some report, more or less casually, contrasting vegetation changes between the middens and their surroundings. However, the floristic contrasts observed by Hrdlička and Meigs did not always appear on dedicated vegetation studies. For example, working in the same coast of British Columbia as this current manuscript, Sawbridge & Bell (1972, *Ecology* 53: 840-849) found that "shell deposits have little influence on composition and structure of forest communities in this area."

To my knowledge, only one study has advanced a similar hypothesis to the one presented in this paper, namely that long-term human occupation does not necessarily imply landscape degradation. In 2014, Vanderplank et al. (*Bioscience* 64(3): 202-209) reported that shell middens in Baja California harbored a unique flora of otherwise rare and endemic species, and concluded that "anthropogenic soils can increase native plant biodiversity and landscape heterogeneity."

What I find is new in this paper is that, while all previous studies on the subject have focused on floristic changes induced by the presence of calcic substrates in shell middens, the current manuscript focuses on productivity changes induced by midden-derived soil calcium.

Data & methodology

To my knowledge, all the statistical methods used in the making of this paper are robust and sound. My only comment is that, because the Methods section is presented at the end, the reader encounters a reference to having used boosted regression trees for data analysis in the caption of Figure 2, before the concept has been introduced in the main text. This forces the reader who is not familiar with the method (many biologists/ecologists are not) to plunge into the Methods to find out what was exactly what was done. But this seems only a minor criticism, and it does not worry me too much.

Conclusions

The main conclusion of the paper is that extensive human use and resource extraction does not necessarily lead to degraded environments and that the coastal First Nations of British Columbia developed shellfish-extraction practices that actually enhanced forest productivity in these calcium-limited ecosystems. In my opinion, the conclusions are robust and perfectly supported by the data.

Suggested improvements

The study is robust and well done, the main points come across quite clearly and I cannot suggest any further improvements.

References

As I mentioned above, the authors are perfectly justified in their claim that their study "is the first

known example of long-term intertidal resource-use enhancing forest productivity." However, they fail to note that their idea has been explored by other authors using not productivity but other ecosystem attributes as indicators of improved ecosystem health, such as the presence of useful, rare, or threatened species, or simply the ability to maintain a higher biodiversity. I think that, without any detriment to this paper, the studies of Hrdlička and Meigs, for example, should be cited as pioneers of this general idea.

Clarity and context

The paper flows well and is clear. Despite having used fairly complex statistical approaches, the main concepts come across very clearly and are easy to follow.

Reviewer #3 (Remarks to the Author):

This is a well-written, interesting article claiming that human activities 1000s of years in the past have positive (not negative) impacts on contemporary forest productivity in the Pacific Northwest. The primary hypothesis is that the harvesting and subsequent dumping of clamshells into coastal middens (i.e., trashheaps) altered soil chemistry by raising pH and increasing calcium, leading to enhanced growth of the dominant tree, western red cedar, which is putatively calcium-deficient.

I have three primary comments on the article, which I hope can be constructively addressed.

First, the link between elevated calcium from shell deposition and enhanced tree growth is tenuous, weakening the proposed mechanism of calcium-enhanced growth and reduced mortality. There is clearly elevated calcium at midden sites, both in the soil and in cedar wood, and these site-level patterns are positively correlated with elevated radial growth and less top-dieback, but there is little mechanistic basis to suggest that cedar trees are calcium-limited. The only citation to this effect in the manuscript is a relatively short brochure from the British Columbia Ministry of Forests (#28), where there is an uncited line stating: "Cedars require lots of calcium, especially for the proper development of their tops. Scientists think the dead tops of these trees are caused by calcium deficiencies in the moist, acidic soils of these areas."

I did a small amount of literature searching hoping to find empirical studies examining calcium-limitation in red cedars. Hopefully the authors can better document this link, but I only found one study that examined tree growth rates (near the same sites as in this study) relative to soil and tissue concentrations of calcium and concluded that cedars "accumulate Ca in excess of its nutrient needs", as there was no relationship between cedar growth and soil Ca, but strong relationships between soil P, K, and terminal growth (Radwan and Harrington 1986, Canadian Journal of Forest Research 16:1069-1075). The findings of Radwan and Harrington suggest that the positive correlation between soil calcium and tree growth are potentially unrelated and driven instead by site P, which is also elevated at habitation sites (Extended Fig 4). Side note: there must be a typo in the last line of the second page of the article, as it claims that P is "similar" across sites, but it is significantly elevated in Extended Fig 4.

Thus, my suggestions here are twofold. If the hypothesis is that the link between cedar health and calcium is direct as postulated, then a better link needs to be made between the two that rules out the covarying influence of P. Second, if the hypothesis instead is that calcium deposition influences the availability and abundance of limiting nutrient P, then the article would be enhanced if it further explored the biogeochemical nature of these interactions. I don't think the article is necessarily weakened by this indirect explanation of calcium altering soil nutrient availability, but as written the article draws a strong relationship between tree productivity and soil calcium while minimizing the

effects of a confounded element.

Second, fire was repeatedly mentioned as another potential driver of site productivity, but the actual basis for this interaction was never discussed. The article could benefit from either fleshing out fire as another driver of site productivity, or ruling it out if it is secondary with respect to shell deposition.

Third, the selection of control sites is critical to ruling out confounding effects of pre-existing conditions, but I found this section to be rather vaguely explained. For example, the authors claim that occupation predates the arrival of both red cedar and modern vegetation communities and thus site selection is unlikely to have driven enhanced cedar growth. However, what if only the most productive sites were chosen for habitation, or if long-lived settlements only persisted in productive sites, and these differences in productivity predated vegetative turnover to cedar? Both scenarios would result in midden presence being confounded with enhanced forest productivity, but there wouldn't necessarily be a direct link between middens and forest growth. This is critical because most of the patterns in this paper are site-level comparing midden to off-midden, implying that any differences are due to midden deposition and not some other unstudied factor. To be fair, the authors do an estimable job of examining other factors like aspect, slope, elevation, and surface material, but even here the results in Extended Table 2 suggests that aspect and mean slope might be more important than distance to midden in explaining canopy height and vegetation greenness, and at least as important for canopy cover. Side note: the authors state that distance from midden explained 58% of variance in forest width, but Extended Table 2 shows that this value should be 38%. Clarification is needed.

Overall, this story is interesting, but the devil is in the details, and I was not convinced that was a compelling reason to believe that the patterns of enhanced cedar growth aren't being driven by naturally elevated phosphorous levels at sites that might have also attracted stable settlements of coastal inhabitants irrespective of subsequent midden deposition, nor was i convinced that other site characteristics weren't stronger drivers. Thus, I suggest revisiting some of these topics with either enhanced remarks about calcium limitation, site selection, and site-level variation in tree-limiting factors, or moderation of the conclusions.

REVIEWERS' COMMENTS:

Reviewer #1 (Remarks to the Author):

Summary of key results: The revision of the manuscript now provides better context of the Coast Range ecosystems and this anomaly to expected results because of historic human activity. I am satisfied with the revisions and recommend acceptance of the manuscript for publication.

Reviewer #2 (Remarks to the Author):

An original version of this paper was reviewed on February 18th, 2016. This second review is basically oriented to check how the authors have responded to the first review comments.

Summary of key results:

The author tests in this study the effect of large archaeological shell middens -accumulated in the coast of British Columbia during the last 6,000 years as a result of shellfish harvesting by the region's First Nations- on the growth of native red cedar forests. Because this region is very wet, with more than 4,000 mm of rainfall that leach cations away from the acidic local soils, calcium is a limiting resource for plant growth. The authors show that red cedar trees established on large archaeological shell middens significantly enhance their growth and increase the productivity of the temperate rainforest.

Original review comments:

In my first review I suggested that the paper should make reference to important early studies that have attempted to evaluate the impact of archaeological middens on current vegetation, such as Hrdlička (1937; *Science* 86: 559-560) and Meigs (1938; *Science* 87: 346), as well as more recent studies such as Vanderplank et al. (2014; *Bioscience* 64(3): 202-209), possibly stressing the fact that, while all these previous studies focused on floristic changes induced by the presence of the calcic substrates derived from ancient shell middens, the current manuscript focuses on productivity changes induced by midden-derived soil calcium. I also stressed the fact that, because the Methods section is presented at the end, the reader encounters a mention to boosted regression trees in the caption of Figure 2 before the concept has been introduced in the main text, forcing the reader to go to the Methods section to find out what was exactly done.

Author's response to review comments:

Both comments have been addressed adequately by the author. He has adequately answered my queries and concerns, and I am satisfied with this new version of the paper.

The paper now flows well and is clear. Despite having used fairly complex statistical approaches, the main concepts come across clearly.

Reviewer #3 (Remarks to the Author):

I have read the response comments and the revised manuscript. In my opinion although the manuscript is improved, the revisions are mostly editorial in nature and generally lacked the clarity and heft provided in the response to the referees. Probably this is due to space limitations, but my primary comment is that the direct versus indirect nature of the calcium-phosphorous-growth linkage is still not quite clearly presented in the revised manuscript.

Below I revisit my original comments. As a side note I thought that the responses to the other referees were fine.

ISSUE 1: Calcium as a limiting element for tree growth

The authors originally claimed that calcium was limiting to red cedar growth, and thus the presence of calcium in middens directly contributed to elevated forest productivity in habitation sites. However, there is little to no literature to back this claim, and in fact enhanced calcium concentration is confounded with elevated phosphorous, which has been identified as a limiting element in red cedar. In the author response, the authors present a different mechanism, whereby calcium might indirectly promote tree growth by raising the pH and thus liberating the limiting element phosphorous. I find this mechanism much more plausible given the status of the background literature, and the authors seem to agree based on their response to the original comments. Nevertheless, it was still somewhat buried in the revised manuscript, presented primarily as background material and not overly discussed as an interpretation of the data.

My suggestions are: First, in my opinion it would help to include the graphs regarding pH and phosphorous in Figure 3 in the manuscript and not the extended data. Extra space can be gained in Figure 3 by deleting the redundant x-axis labels, resulting in 3 stacked panels on the left and right surrounding the tree picture. Second, I would present this interpretation in the manuscript the exact same way it is in the author response (P is limiting but Ca liberates it by raising pH); it seemed glossed over in the manuscript by presenting it mainly as Introduction material rather than a true discussion point. Sadly, the MS lacks line numbers or I would point to specific sections where it could be added, and the authors could have similarly pointed to sections that had been revised.

Without fixing this crucial part of the story the reader will assume that calcium is a limiting element to tree growth, but it appears to only indirectly influence growth and productivity. This doesn't hurt the story in my opinion, but rather adds context. In effect, Native Americans accidentally limed the forest and actually made it grow faster and better. Cool!

ISSUE 2: inclusion of fire

Here again I thought fire was addressed somewhat peripherally in the original manuscript, but mostly the revisions simply add the observation that elevated calcium content and fire are covarying at the habitation sites. The authors mention in a few places that charcoal from ancient fire can alter pH and thus lead to the liberated critical elements, but I would suggest a reference or two to support this claim.

Additionally, I was wondering whether the presence and thus the impact of fire extended across the entire midden, or could the two be separated at the local, within-midden scale, essentially by comparing tree growth within a fire-belt versus outside of a fire belt? This would certainly help to tease apart the influence of these variables.

ISSUE 3: selection of control sites.

I am fine with the comments provided by the authors, although I am still somewhat confused by the all versus control versus habitation sites in the figures (see also below).

OTHER COMMENTS

Figure 2 legend should better explain the differences between sites labeled as 'all', 'control', versus 'habitation'. As is, control sites are best explained in the legend for Figure 3, and it was unclear what the 'all' sites are.

Reviewer #1

Summary: The manuscript in review provides an interesting side-story about historical human inputs of calcium via accumulation of shell middens and the slow degradation that provided a steady leaching of Ca, which reduced the Ca deficiency otherwise seen in these soils. The normal Ca depletion has led to die-back of the tops of tall trees in the region, but not where the deficiency was counteracted by the slow additions of Ca from the middens. This story is interesting in that it reminds us to understand the historic land use effects on current forest structure and function, particularly in long-lived forests like the Coast Range of western N America.

Originality: The paper isn't original in terms of the N-Ca coupling, but it has an interesting exception to the rule when historic coastal human habitation alters this coupling via effects on Ca. It is well-known that high nitrogen (N) accumulation in Coast Range forests of Oregon, Washington and British Columbia acidified soils and depleted calcium. A recent study determined that biotic conservation and tight recycling of Ca increased in response to whole-ecosystem Ca depletion, which was indicated by preferential accumulation of Ca in biomass and surface soil. i.e. coupled N-Ca cycling under long-term soil N enrichment.

Methods, statistics: The study design and statistics are appropriate, with paired plots (with and without shell middens), which resulted in finding taller canopies, higher productivity, and less top die-back associated with middens. In other studies, Ca pools in aboveground biomass decreased significantly as soil N increased (e.g. Perakis et al. 2013), whereas this new study shows that western redcedar grown on middens showed significantly higher wood calcium levels. That is, the results are consistent with previous studies along nitrogen gradients in the Coast Range.

Conclusions, clarity and content: The paper stands alone on this topic. The justification for the importance of this finding is weak. I don't think it is useful to suggest that this study challenges the assumption that human activities degrade forests, such as by commercial logging and other resource extraction. These are completely different activities, and they have been shown to degrade forests. It sounds as if the authors are trying to say that if the middens are good for forests, so are all of these other activities, when I don't think that is their intent.

***Author response:** Thanks for this careful review. You are correct that our statement regarding 'challenging the assumption' could be interpreted as our wanting to reconsider how all human activities (e.g. logging) should be thought of. And as you rightly suggest, this is not our intention. Instead, we have modified the text in the first paragraph of the main text to read: "...which provide alternative models to human presence resulting in environmental degradation." We hope that this clarifies our position on the issue and we welcome any additional feedback.*

Rather, simply state that historical *human activity in coastal areas can result in pockets of higher forest productivity associated with the slow leaching of shell middens and fish bones than might otherwise be predicted for high soil N and depleted Ca. Coastal areas, rivers and lakes are obvious*

places where historic human habitation and consumption of fish and shellfish could leave their marks on the soil fertility and forest productivity that are different from the surrounding forests.

Author response: *Thanks for the great ideas. In the concluding paragraph, we have incorporated your suggestions and modified our sentence to: “Here we offer alternative consequences of extensive and long-term human management in coastal areas. Pockets of enhanced forest productivity associated with the slow leaching of shell middens and the influence of past fires than might otherwise be predicted for coastal soils with high nitrogen and low availability of other important nutrients such as calcium (Perakis et al 2013).*

The paper is publishable in Nature once revisions improve this part of the paper and the citations.

Suggested Citation: Perakis, S. et al. 2013. Forest calcium depletion and biotic retention along a soil nitrogen gradient. *Ecological Applications* 23:1947-61.

Author response: *Thank you very much for your thoughtful feedback and constructive editorial suggestions. We added Perakis et al (2013), which strengthens both our argument and it also does an excellent job providing addition regional context.*

Reviewer #2

Summary of key results

The author tests in this study the effect of large archaeological shell middens -accumulated in the coast of British Columbia during the last 6,000 years as a result of shellfish harvesting by the region's First Nations- on the growth of native red cedar forests. Because this region is very wet, with more than 4,000 mm of rainfall that leach cations away from the acidic local soils, calcium is a limiting resource for plant growth. The authors show that red cedar trees established on large archaeological shell middens significantly enhance their growth and increase the productivity of the temperate rainforest.

Originality and interest

The most interesting aspect of this study is the finding that long-term occupation by humans and large scale accumulation of human-created rubble in the form of shell middens is actually contributing in present times to significantly increase the productivity of the coastal temperate rainforest. This finding, the author argues, contradicts the generalized notion that long-term human occupation leads to the formation of degraded landscapes.

Other studies have attempted to evaluate the impact of archaeological middens on current vegetation. In 1937, Hrdlička (Science 86: 559-560) reported that shell middens in Alaska "present wide and in cases seemingly almost absolute botanical differences from the rest of their region." In 1938, Meigs (Science 87: 346) noted that middens in Lower California harbored different species and looked greener and denser than the surrounding desert vegetation. Many studies archaeological studies have been done on the shell middens of the Pacific coast of the Americas, and some report, more or less casually, contrasting vegetation changes between the middens and their surroundings. However, the floristic contrasts observed by Hrdlička and Meigs did not always appear on dedicated vegetation studies. For example, working in the same coast of British Columbia as this current manuscript, Sawbridge & Bell (1972, Ecology 53: 840-849) found that "shell deposits have little influence on composition and structure of forest communities in this area."

***Author response:** Thanks for bringing up these references (Hrdlička 1937 and Meigs 1938). We were not aware of these but agree that they strengthen the evidence for the response that we have documented. They have both been included. Regarding Sawbridge & Bell (1972), our study doesn't address forest composition but rather structure (through productivity and growth metrics). Furthermore, Sawbridge & Bell didn't actually compare vegetation structure/composition to non-midden (ie. control) sites, despite this being reported in the abstract. Any comparisons, as far as we can tell, were not done quantitatively.*

To my knowledge, only one study has advanced a similar hypothesis to the one presented in this paper, namely that long-term human occupation does not necessarily imply landscape degradation. In 2014, Vanderplank et al. (Bioscience 64(3): 202-209) reported that shell middens in Baja California harbored a unique flora of otherwise rare and endemic species, and concluded that "anthropogenic soils can increase native plant biodiversity and landscape heterogeneity."

What I find is new in this paper is that, while all previous studies on the subject have focused on floristic changes induced by the presence of calcic substrates in shell middens, the current manuscript focuses on productivity changes induced by midden-derived soil calcium.

Data & methodology

To my knowledge, all the statistical methods used in the making of this paper are robust and sound. My only comment is that, because the Methods section is presented at the end, the reader encounters a reference to having used boosted regression trees for data analysis in the caption of Figure 2, before the concept has been introduced in the main text. This forces the reader who is not familiar with the method (many biologists/ecologists are not) to plunge into the Methods to find out what was exactly what was done. But this seems only a minor criticism, and it does not worry me too much.

***Author response:** Thanks for the comments and great contextualization of our study. That was great to read. Your comment about the technical portion of the Fig. 2 caption is excellent and something that we hadn't thought of. We added the following to the figure caption: "...standard deviation (red and blue ribbon) of bootstrapped boosted regression tree models, which are an additive regression model where terms are simple trees that are fitted in a forward, stagewise fashion (Elith et al. 2008), and mean forest width across entire study area..."*

Does this addition help clarify the methods from the figure caption? We are very interested to hear your thoughts on this.

Conclusions

The main conclusion of the paper is that extensive human use and resource extraction does not necessarily lead to degraded environments and that the coastal First Nations of British Columbia developed shellfish-extraction practices that actually enhanced forest productivity in these calcium-limited ecosystems. In my opinion, the conclusions are robust and perfectly supported by the data.

Suggested improvements

The study is robust and well done, the main points come across quite clearly and I cannot suggest any further improvements.

References

As I mentioned above, the authors are perfectly justified in their claim that their study "is the first known example of long-term intertidal resource-use enhancing forest productivity."

However, they fail to note that their idea has been explored by other authors using not productivity but other ecosystem attributes as indicators of improved ecosystem health, such as the presence of useful, rare, or threatened species, or simply the ability to maintain a higher biodiversity. I think that, without any detriment to this paper, the studies of Hrdlička and Meigs, for example, should be cited as pioneers of this general idea.

Author response: We agree and on your advice we have added reference to the work of Hrdlička (1937) and Meigs (1938).

Clarity and context

The paper flows well and is clear. Despite having used fairly complex statistical approaches, the main concepts come across very clearly and are easy to follow.

Line edits:

Replace 'x' with multiplication sign

Author response: This was corrected throughout the manuscript

Correct LAT LONG

Author response: Correction made

Italicize 'n' in (n=X) and 't' in t-test

Author response: Corrections made throughout

Reviewer #3

This is a well-written, interesting article claiming that human activities 1000s of years in the past have positive (not negative) impacts on contemporary forest productivity in the Pacific Northwest. The primary hypothesis is that the harvesting and subsequent dumping of clamshells into coastal middens (i.e., trashheaps) altered soil chemistry by raising pH and increasing calcium, leading to enhanced growth of the dominant tree, western red cedar, which is putatively calcium-deficient. I have three primary comments on the article, which I hope can be constructively addressed. First, the link between elevated calcium from shell deposition and enhanced tree growth is tenuous, weakening the proposed mechanism of calcium-enhanced growth and reduced mortality. There is clearly elevated calcium at midden sites, both in the soil and in cedar wood, and these site-level patterns are positively correlated with elevated radial growth and less top-dieback, but there is little mechanistic basis to suggest that cedar trees are calcium-limited. The only citation to this effect in the manuscript is a relatively short brochure from the British Columbia Ministry of Forests (#28), where there is an uncited line stating: "Cedars require lots of calcium, especially for the proper development of their tops. Scientists think the dead tops of these trees are caused by calcium deficiencies in the moist, acidic soils of these areas."

I did a small amount of literature searching hoping to find empirical studies examining calcium-limitation in red cedars. Hopefully the authors can better document this link, but I only found one study that examined tree growth rates (near the same sites as in this study) relative to soil and tissue concentrations of calcium and concluded that cedars "accumulate Ca in excess of its nutrient needs", as there was no relationship between cedar growth and soil Ca, but strong relationships between soil P, K, and terminal growth (Radwan and Harrington 1986, Canadian Journal of Forest Research 16:1069-1075). The findings of Radwan and Harrington suggest that the positive correlation between soil calcium and tree growth are potentially unrelated and driven instead by site P, which is also elevated at habitation sites (Extended Fig 4).

***Author response:** These are really great suggestions and we believe have improved the paper in significant ways. They have made us rethink some of our interpretations. Thank you. We have addressed these suggestions and incorporated new sections in the manuscript in a number of ways, and present the details below. In particular, please see the comment below on how we incorporated the linkages between Ca/P/fire. We have included reference to Radwan and Harrington's 1986 paper when discussing calcium limitations.*

Side note: there must be a typo in the last line of the second page of the article, as it claims that P is "similar" across sites, but it is significantly elevated in Extended Fig 4.

***Author response:** Thank you for noticing this typo. The text now reflects that indeed, phosphorous was significantly higher at midden sites.*

Thus, my suggestions here are twofold. If the hypothesis is that the link between cedar health and calcium is direct as postulated, then a better link needs to be made between the two that rules out the

covarying influence of P. Second, if the hypothesis instead is that calcium deposition influences the availability and abundance of limiting nutrient P, then the article would be enhanced if it further explored the biogeochemical nature of these interactions. I don't think the article is necessarily weakened by this indirect explanation of calcium altering soil nutrient availability, but as written the article draws a strong relationship between tree productivity and soil calcium while minimizing the effects of a confounded element.

***Author response:** Thank you for raising this concern and after going through your comments and the manuscript, we agree with your comments that the basis for the interaction of fire and site productivity wasn't fleshed out as clearly as we thought. Also, rather than thinking of it as primary and secondary, we prefer to consider these as direct (midden) and indirect (changes in pH and soil properties through addition of CaCO₃ and fire) but we believe that the sentiment is the same.*

Based on your suggestions, we have clarified and elaborated on a number of issues pertaining to calcium, phosphorous and fire. In doing so, we are confident that the linkages between these factors is clearer to the reader and that this offers a more accurate and compelling story. We have expanded our discussion of calcium and now identify important linkages between calcium, pH, phosphorous and fire. Acidic soils, which are common on the British Columbia coast, reduce the availability of essential nutrients such as phosphorous. The addition of calcium and charcoal to the habitation sites increases the pH (see Extended data figure 4), which in turn, makes phosphorous more readily available to plants. In addition to this, archaeological sites containing history of fire (e.g., hearths) have been shown to have higher levels of phosphorous (Holliday and Gartner 2007), resulting from either constituents of the charcoal itself or from physical modifications to the soil that enhance nutrient cycling (Lehmann et al. 2011). Thus, the increased productivity that we report at habitation sites results from the addition of calcium and charcoal, which then lead to higher levels of available phosphorous. Based on your suggestion, we have revisited and reworked our conclusions related to direct linkages between calcium and forest health. Instead, we focus on discussing greater linkages between shell middens, fire and forest productivity. These linkages with pH, Ca, and P have been incorporated throughout the manuscript.

We would like your opinion as whether to keep the P and pH figure in the extended data Fig. 4 or whether it would be better to make it an addition figure in the manuscript?

Second, fire was repeatedly mentioned as another potential driver of site productivity, but the actual basis for this interaction was never discussed. The article could benefit from either fleshing out fire as another driver of site productivity, or ruling it out if it is secondary with respect to shell deposition.

***Author response:** Thank you for these excellent suggestions: we did not do a good enough job in incorporating our fire data. The manuscript now includes discussion on the linkages*

between fire, phosphorous, and calcium and ultimately how they influence productivity (see the above response for details). Fire offers two potential mechanisms for enhancing productivity. First, by increasing the pH of the soil and secondly, by the bulk addition of nutrients contained within the charcoal itself. We under-reported on this in the first version of our manuscript as we were unsure how best to pursue it, and thank you for suggesting we A) do more with this, and B) offering an excellent avenue to enhance our story. This has made the manuscript much clearer and stronger.

Third, the selection of control sites is critical to ruling out confounding effects of pre-existing conditions, but I found this section to be rather vaguely explained. For example, the authors claim that occupation predates the arrival of both red cedar and modern vegetation communities and thus site selection is unlikely to have driven enhanced cedar growth. However, what if only the most productive sites were chosen for habitation, or if long-lived settlements only persisted in productive sites, and these differences in productivity predated vegetative turnover to cedar? Both scenarios would result in midden presence being confounded with enhanced forest productivity, but there wouldn't necessarily be a direct link between middens and forest growth. This is critical because most of the patterns in this paper are site-level comparing midden to off-midden, implying that any differences are due to midden deposition and not some other unstudied factor. To be fair, the authors do an estimable job of examining other factors like aspect, slope, elevation, and surface material, but even here the results in Extended Table 2 suggests that aspect and mean slope might be more important than distance to midden in explaining canopy height and vegetation greenness, and at least as important for canopy cover.

Author response: *While it isn't possible to specifically know the presettlement levels of productivity, we are confident that the differences in productivity that we report here can be attributed to the human-modifications to the physical and biochemical components of the site. With shell middens changing the slope, drainage, pH, calcium, and in conjunction with fire, the phosphorous levels, we feel confident that this exceeds the influence of pre-settlement conditions. At the least, it would have enhanced those already productive conditions. Lastly, the control sites that we selected for this study were the most productive sites that we could find that do not have shell middens. There are no local differences in bedrock geology (it is all underlain by the same granodiorite rock with local surficial deposits of tills). Banner et al (2005), state 'The scarcity of glacial till in this coastal environment highlights the importance of bedrock geology. Most soils develop directly from the weathering of bedrock or colluvial material. This contrasts with many other areas where a mantle of glacial till of mixed lithology masks the influence of bedrock. In addition, sharp contrasts in bedrock type occur on the outer coast, from the hard, slowly weathering granodiorites with relatively low amounts of available nutrient elements, to the much softer, easily weathered metamorphic rocks and limestone with more nutrient rich lithologies.' Thus, the results we report are conservative in that comparisons are made to the most productive sites with similar underlying geology and not just to random sites along the coastline.*

Banner, A., LePage, P., Moran, J. & de Groot, A. The HyP 3 Project: pattern, process, and productivity in hypermaritime forests of coastal British Columbia—a synthesis of 7-year results. BC Min. For., Res. Br., Victoria, BC Spec. Rep 10, 142 (2005).

Also, we added this sentence to the paper in the study site section of the methods, ‘Compared to inland areas of the British Columbia coast, Calvert and Hecate Island are comprised of homogenous and nutrient-poor quartz granodiorite bedrock geology (Banner et al 2003).’

Your point about long-lived settlements only persisting in productive sites is very interesting and something that we now realize was not suitably addressed in the manuscript. The habitation sites included in this study have a variety of occupation histories (see Extended Data Table 1: Continuous habitation & depth to midden). Thus, our data include sites that may have been abandoned for the reasons you bring up and yet the eco-cultural legacy persists at these sites due to the discussed modifications. If the sites were selected for initial higher levels of productivity, and not other important factors like access to ocean, slope of shoreline, degree of protection from weather, then we would expect more evidence of continuous occupation – but that isn’t the pattern that we see. The reasons for differing lengths of occupation are complex and at this stage in the archeological research, not well understood.

To clarify these points, we have added additional information in the manuscript where site selection criteria were explained: “...were compared to control sites that were selected for having high productivity, similar forest structure and site attributes (aspect, slope, proximately to shore) but lacked any above or below ground evidence of shell midden (Fig. 1) and thus are considered not to be habitation sites.”

Side note: the authors state that distance from midden explained 58% of variance in forest width, but Extended Table 2 shows that this value should be 38%? Clarification is needed.

Author response: *Thanks for noticing this. The correct value is 38% and the text now reflects this change.*

Overall, this story is interesting, but the devil is in the details, and I was not convinced that was a compelling reason to believe that the patterns of enhanced cedar growth aren't being driven by naturally elevated phosphorous levels at sites that might have also attracted stable settlements of coastal inhabitants irrespective of subsequent midden deposition, nor was i convinced that other site characteristics weren't stronger drivers. Thus, I suggest revisiting some of these topics with either enhanced remarks about calcium limitation, site selection, and site-level variation in tree-limiting factors, or moderation of the conclusions.

Author response: *The manuscript now incorporates greater linkages between shell middens, fire and forest productivity. The main difference is the discussion of pH and how the addition of both calcium carbonate (through shell deposition) and charcoal (from fire) increase the*

soil pH. This increase in soil pH, as well as the additional nutrients from shell midden and charcoal, results in significantly elevated phosphorous levels – all of which may contribute to higher productivity. Regarding the possibility of naturally elevated phosphorous levels, we stated in an earlier comment that all sites are underlain by the same granodiorite bedrock, which results in soils with naturally low nutrient levels (and similar between sites) and add this important information to the study sites section of the manuscript. In terms of site selection, we have added additional information about the criteria used and additional information regarding your ideas raised about site abandonment and pre-occupation site history (as described above). Thank you for these suggested revisions: they have helped us greatly improve our manuscript.